# MYC-Induced Replicative Stress: A Double-Edged Sword for Cancer Development and Treatment

**DOI:** 10.3390/ijms22126168

**Published:** 2021-06-08

**Authors:** Laura Curti, Stefano Campaner

**Affiliations:** Center for Genomic Science of IIT@CGS, Fondazione Istituto Italiano di Tecnologia (IIT), 20139 Milan, Italy

**Keywords:** MYC, DNA replication, transcription, replicative stress, transcription stress, cancer therapy

## Abstract

MYC is a transcription factor that controls the expression of a large fraction of cellular genes linked to cell cycle progression, metabolism and differentiation. MYC deregulation in tumors leads to its pervasive genome-wide binding of both promoters and distal regulatory regions, associated with selective transcriptional control of a large fraction of cellular genes. This pairs with alterations of cell cycle control which drive anticipated S-phase entry and reshape the DNA-replication landscape. Under these circumstances, the fine tuning of DNA replication and transcription becomes critical and may pose an intrinsic liability in MYC-overexpressing cancer cells. Here, we will review the current understanding of how MYC controls DNA and RNA synthesis, discuss evidence of replicative and transcriptional stress induced by MYC and summarize preclinical data supporting the therapeutic potential of triggering replicative stress in MYC-driven tumors.

## 1. MYC Is a Transcription Factor and Oncogene

c-MYC (henceforth MYC), which is a transcription factor belonging to the basic helix-loop-helix–leucine zipper (bHLH-LZ) family, requires dimerization with its obligatory partner MAX in order to regulate transcription. *MYC* was initially identified as an avian viral oncogene (*v-MYC*) responsible for myelocytomatosis in the chicken, a condition leading to the development of tumors, primarily sarcomas and lymphomas [1,2]. Shortly after its discovery as a viral oncogene, its cellular counterpart (*c-MYC*), as well as its oncogenic genomic rearrangements in human tumors, were identified [3]. Subsequently, two *MYC* paralogues, *MYCN* (initially identified in neuroblastomas) and *MYCL*, were cloned in neuroblastoma and lung cancers, respectively. Both proteins share the same cellular functions of MYC, with a less ubiquitous, more tissue-specific expression pattern in humans.

In physiological conditions, *MYC* is part of the immediate-early gene program, its expression being controlled by cytokines and growth factor signaling, where it regulates the transcription of genes required for cell division, cell growth and cellular metabolism [4].

Post-translationally, MYC activity is mainly regulated by a network of bHLH-LZ proteins that can act either as agonists or antagonists of MYC [5]. Within this network, the formation of the MYC/MAX heterodimer is directly challenged by the MGA and MXDs proteins which compete with MYC for binding to MAX. Thus, both MGA and MXDs act as MYC antagonists. A further layer of the network has a central node in the MLX protein, which is a shared partner for either MXDs or the MLXIP/MLXIPL proteins. While MLX can be considered a MYC agonist, by virtue of its ability to titrate MXDs away from MAX, the MLXIP/MLXIPL proteins, by preventing MLX association to MXDs, are indirect antagonists of MYC. This “Extended MYC network” is crucial to balance cell growth and metabolism during cell differentiation and homeostatic growth, both in physiological and pathological conditions [5].

In cancer, MYC is frequently expressed at high levels and in a constitutive way, as a result of either direct genomic rearrangements of the *MYC* locus or as a consequence of the aberrant activation of upstream oncogenic pathways, which support constitutive and robust transcriptional regulation of gene programs essential for tumor initiation and growth. Several preclinical studies have highlighted potent oncogene addiction in MYC-driven tumors [6], and loss of function analysis in several tumor models (not MYC driven) has suggested a general strong dependency of tumor cells on MYC activity [7]. These observations have raised a strong interest in identifying therapeutic strategies based on MYC inhibition in cancer cells. Solutions that concern either direct inhibition of MYC activity by small molecules or biologicals (i.e., OMOMYC-based therapies) or the targeting of genetic dependencies established by hyper-activation of MYC are currently under development [7].

## 2. The Landscape of MYC-Dependent Transcription

MYC controls the transcription of a large number of protein-coding genes and non-coding RNAs, being both an activator or a repressor, depending on the specific target. Over the years, it has become increasingly clear that MYC regulates genes generally linked to cell cycle, metabolism and differentiation. While the ontology of MYC targets may be conserved across cell types and in different organisms, the precise identity of these genes varies in a context-dependent way. This owes to (i) the strong influence of accessibility/epigenetic bookmarking in determining MYC binding to genomic loci and (ii) the evidence that MYC-dependent transactivation scales with its expression levels, both in terms of mRNA synthesis rate and, more relevantly, in the number of MYC-regulated genes [4,8].

The potential for widespread control of transcription by MYC, based on the initial interpretation of genomic data, has stirred considerable interest in understanding whether MYC would regulate gene expression in a sequence-specific way or rather as a general transcription factor. As detailed in the following paragraphs, there is now sufficient evidence combining genome-wide chromatin studies, expressional analysis and biochemical data, which allows the formulation of a unifying model that supports selective transcriptional regulation by MYC and accounts for its widespread transcriptional activity in cancer cells [4,8]. Like other bHLH-LZ transcription factors, the MYC/MAX dimer lacks pioneering activity and relies on accessible chromatin in order to scan the genome for its target sites. Initial MYC/MAX binding to its target sites is dictated by binding to the DNA backbone and protein–protein interactions. This is favored by a permissive chromatin state, characterized by both histone acetylation (H3K27Ac) and methylation (H3K4me3 at promoters and H3K4me1 at enhancers) [4,9].

Then, the proper positioning of the MYC/MAX dimer on the cognate sequence recognition site(s) on DNA (i.e., the E-box sequence), triggers gene transactivation by promoting the recruitment of RNA polymerases and by releasing the paused polymerases from promoters [4]. Sequence-dependent binding may also be a requisite for further MYC-dependent epigenetic bookmarking of promoters and enhancers, which is invariably associated with MYC binding [4,8,9]. In physiological conditions (i.e., low MYC expression), MYC binding is mainly restricted to a subset of promoters and results in selective regulation of genes with canonical E-box promoters [10,11]. During oncogenic transformation, when MYC levels are raised due to either genetic events or oncogenic signaling, MYC will bind an increasing number of promoters and enhancers, and depending on the level of expression, may reach a stage at which it binds all accessible regulatory regions (a phenomenon dubbed as MYC invasion) [11,12,13,14]. In these extreme cases, the number of MYC-regulated genes can rise dramatically; yet, both binding affinity and efficiency of transactivation will still be modulated by the presence of E-boxes (i.e., sequence-specific gene regulation) [11,13]. The prominent increase in enhancers bound by MYC may underscore the relevance of distal regulatory elements in cancer cells, whereby MYC drives their epigenetic remodeling by stimulating the deposition of chromatin-activating marks [11] and may favor promoter–enhancer contacts [15].

## 3. Structure and Function Insights

Several biochemical studies have contributed to the structure–function insight on MYC and the MYC/MAX heterodimer [16]. The bHLH-LZ domain located at the C-terminus of MYC mediates both its dimerization with MAX and the sequence-specific binding to DNA. In particular, the C-terminal leucine zipper and Helix-2 of the MYC’s HLH motif stabilize the heterodimer and impose topological constraints that prevent the formation of MYC homodimers. The Helix-1 of the two HLHs of the MYC/MAX complex contributes to the stabilization of the dimer and positions the adjacent basic region in an orientation that allows proper insertion of this positively charged sequence on both sides of the DNA major groove. Within the basic regions of MYC and MAX, specific residues contribute to the binding of the DNA phosphodiester backbone and the interaction with DNA nitrogen bases, thus accounting for DNA affinity and sequence-specific recognition of the E-box [16].

Studies by site-directed mutagenesis combined with genome-wide chromatin binding and expression analyses have formally demonstrated that arginine 364, 366 and 367, which make contact with the DNA backbone, are the major determinants that account for MYC chromatin interaction, since their mutation completely abolished DNA binding and transactivation [17].

On the other hand, the MYC^HE^ mutant, which bears mutations in base-recognition residues (H359 and E363), displays chromatin affinity and genome-wide distribution undistinguishable from that of wild-type MYC. This suggests that binding of the DNA backbone and the epigenetic state of these loci are sufficient to direct MYC on its genomic targets. Yet, while MYC^HE^ binds the same genomic loci of wild-type MYC, it lacks the fine positioning on the cognate sequence binding motifs, the E-box sequence. A crucial consequence of such loose positioning is that the MYC^HE^ mutant lacks proper transactivation of MYC target genes, since a consistent portion of these are either not regulated or regulated in the opposite direction. Overall, these analyses support a model in which MYC binding to chromatin is mainly determined by accessibility, while fruitful transactivation needs sequence-specific positioning of MYC on the E-box. This reconciles the genome invasion phenotype observed upon MYC overexpression (which is mainly driven by non-sequence-specific binding), with the persistence of selective (and sequence-specific) transactivation observed in tumor cells [4,8,17].

While the bHLH-LZ domains constitute the entire moiety of MAX, MYC has an extended N-terminal portion that endows the MYC/MAX heterodimer with transcriptional activity and is the target of post-translational modifications and regulations [16].

This N-terminus of MYC contains the transactivation domain and six regions, named MYC homology boxes (MB-0 to MB-V), that are conserved in all the three members of the MYC family (MYC, MYCN and MYCL) [3]. These MYC homology boxes are likely to have a functional role and mediate the interaction of MYC with co-activators and upstream regulators. Recent profiling of the MYC interactome has allowed thorough mapping of around 300 high-confidence MYC interactors, and defined the structural relevance of MYC homology boxes in mediating recruitment of such factors for gene modulation [18]. This analysis has revealed extensive interaction of MYC with factors associated with the three RNA polymerases (RNA pol I, II and III) as well as with numerous transcription factors, complexes involved in epigenetic modifications such as histone acetylation (STAGA, TIP60 and ATAC complex), histone methylation (COMPAX complex) and nucleosome remodeling (mostly SWI/SNF-related), which account for positive regulation of gene expression. This analysis also highlighted the association of MYC with histone deacetylases (SIN3A and BHC complex) and non-canonical PRC1 components (i.e., co-factors that associate with the core components of the PRC1 complex), which are possibly linked to gene repression. The same study showed MYC interaction with several RNA processing factors and spliceosome components, as well as with proteins involved in ribosome biogenesis. This observation may perhaps suggests that MYC may regulate co-transcriptional processing of its target genes to regulate splicing of nascent RNA or to foster the inclusion of newly transcribed RNA into ribonucleoprotein complexes.

Intriguingly, several DNA replication factors were also identified, possibly supporting a role for MYC in the control of DNA replication or perhaps underscoring the proximity of MYC-dependent transcriptional complexes with the DNA replication machinery, which may indicate an intrinsic propensity for potential conflict between these two processes.

## 4. Activation of Transcription by MYC

Although MYC activates genes transcribed by RNA pol I, RNA pol II and RNA pol III, to date most of our understanding concerning how MYC transactivates is based on the analysis of RNA pol II-regulated genes. In general, MYC transactivation is a multi-layered process which entails interaction and recruitment of factors involved in nucleosome remodeling, chromatin modifications and its direct interaction with basal RNA pol II-associated factors [4] (Figure 1). This favors the recruitment of the RNA polymerase at promoters, triggers its pause release and supports processive elongation. The recruitment of histone acetyltransferase complexes (HATs) accounts for the increase in histone acetylation at promoters and enhancers that are activated by MYC. Among these HATs, the STAGA and the TIP60 complex are tethered to MYC-box II via binding of TRAPP, a subunit shared among these two multimeric HATs [18,19,20]. Their recruitment is essential for MYC-dependent transformation, but dispensable for the activation of physiological MYC targets, suggesting that MYC induced epigenetic remodeling is a limiting step in tumor development [18,21]. MYC interacts also with WDR5 [22,23], a ubiquitous component of MLL methyltransferase complexes. This interaction, which is mediated by MYC-box IIIb, is essential for MYC to bind to and regulate a set of genes linked to biomass accumulation. Disruption of the MYC–WDR5 interaction in preclinical cancer models promotes widespread apoptosis and tumor collapse [23]. Several lines of evidences connect MYC to factors that regulate RNA pol II activity, accounting for both recruitment of the polymerase at promoters and its activation. MYC binds the TATA-binding protein (TBP) [24], an essential component of the transcription initiation complex TFIID, though residues 98–111. TFIID is a multimeric protein complex of thirteen TBP-associated factors, which is responsible for promoter recognition and the assembly of the pre-initiation complex at the transcriptional start site (TSS). Of note, this protein–protein interaction between MYC and TBP may suggest a mechanism for TBP recruitment on TATA-less MYC targets.

Transcription initiation, RNA pol II promoter escape and processive elongation are regulated by the interaction of MYC (through MB-0) with several basal transcription factors, such as components of the TFIIF, the PAF1c, pTEFb and NELF complexes. Similarly to MYC-box II, mutations in MYC-box 0 affect MYC-induced transformation [18].

Transcriptional initiation, which is marked by the phosphorylation of Ser5 of the RNA pol II C-terminal domain, is rate limited by MYC-dependent recruitment of SPT5/SPT6, the two components of DSIF [25]. This facilitates the CDK7-dependent assembly of RNA pol II–DSIF complexes and, by generating a substrate for pTEFb, is required for the subsequent conversion of DSIF from a pause factor to an elongation factor.

The transition from initiating to elongating RNA pol II (promoter release) requires also the MYC-dependent recruitment of the PAF1c complex and the ubiquitin-mediated turnover of promoter-associated MYC, which favors the transfer of PAF1c from MYC to RNA pol II, thus triggering promoter escape and processive elongation [26]. Since loss of PAF1c, while altering RNA pol II dynamics, has a limited impact on transcription, it is possible that this molecular circuit might be required to mitigate or repress intrinsic transcriptional stress [26].

Besides acting on the promoter-proximal control of gene expression, MYC is also able to engage enhancers. A key factor linking the epigenetic priming of enhancers to the release of promoter-paused RNA pol II is the chromatin reader BRD4 which, by binding MYC and acetylated chromatin, fosters enhancer–promoter contacts [27,28] and the recruitment of pTEFb, a key factor that triggers RNA pol II escape from stalled promoters [29,30,31]. We posit that this may be particularly relevant in cancer cells, where enhancers’ invasion by MYC may orchestrate pervasive oncogene-driven transcription [11,31] and 3D-chromatin remodeling [15].

## 5. Transcriptional Repression by MYC

MYC also represses gene transcription, both in in vitro and in vivo models, yet understanding its mechanism of gene repression has proven challenging, and to date, this is only partially explained. Historically, MYC-dependent repression has been linked to binding to other transcription factors such as SP1, SMAD2/3, MIZ1, NF-Y and YY1 [4]. The identification of the interaction of MYC with histone deacetylase complexes (SIN3a and BHC) and non-canonical components of the PRC-1 complex may suggest that, at least in some cases, MYC represses transcription by epigenetic mechanisms, thus favoring the removal of activating histone modifications (i.e., acetylation of histones) or the deposition of repressive chromatin marks (i.e., monoubiquitylation of histone H2A at lysine 119) [18].

With the exception of MIZ1, the best characterized MYC-dependent repressor, the paucity of genomic data has so far prevented the drafting of a general model for MYC-dependent gene repression.

MIZ1 regulates a small subset of genes mainly involved in vesicles trafficking and autophagy; its binding to MYC reshuffles MIZ1 on a subset of MYC-bound genes. On these genes, the transcriptional outcome varies depending on the stoichiometry of the MYC/MIZ1 complex associated with chromatin, with preferential repression at genes with an MYC/MIZ1 ratio close to one [13]. Loss of function experiments conducted in *Miz1 KO* cells or by overexpression of MYC^V394D^, a mutant lacking MIZ1 binding, collectively showed that 20–40% of MYC repressed genes are downregulated in an MIZ1-dependent way [13]. These genes are mainly linked to cell cycle control, self-renewal, apoptosis and cell adhesion. The impairment of cellular differentiation, which is a major consequence of oncogenic deregulation of MYC, is only in part mediated by MIZ1 and is most probably mostly due to the selective repression of lineage-determining transcription factor [32].

Besides direct recruitment of repressive complexes on target genes, repression by MYC can also be a consequence of the broad landscape of its genomic targets, that, even in conditions of overexpression, may limit MYC activity at lower-affinity sites. In this scenario, although MYC overexpression leads to a global increment in its binding to genomic loci, those sites that gain less MYC are associated with gene repression, while those which gain the most are associated with gene activation. This “MYC share model” probably reflects the limiting amount of RNA pol II and/or its associated co-factors, that upon MYC overexpression may not be efficiently recruited on genes bound by MYC with low affinity [33]. Alternatively, MYC may also sequester RNA pol II-associated factors, such as SPT5, into non-functional complexes, thus indirectly dampening the expression of potentially growth-suppressive genes in tumors [25].

## 6. MYC Controls DNA Replication and Replicative Stress

MYC exerts a pleiotropic transcriptional control over DNA replication, both in physiological and in pathological conditions (Figure 2). It controls the G1/S transition and the maintenance of the cell cycle by regulating the expression of cyclins, cyclin-dependent kinases (CDKs) and cyclin-dependent kinase inhibitors (CKIs) [34,35]. In addition, MYC manages the high flux of metabolites required for DNA replication by regulating the expression of the majority of the genes involved in purine and pyrimidine biosynthesis [36,37], and regulates the expression of several genes involved in DNA replication [4].

Several observations indicate that MYC can also control DNA replication in a transcription-independent fashion. Initial evidence began to emerge in the late 1980s, when a direct interaction between MYC and components of the DNA replication machinery was first described [38,39,40,41]. Subsequently, several reports showed a direct interaction between MYC and components of the pre-replication complex (pre-RC); MYC was shown to localize to DNA replication origins and to promote pre-RC assembly by physically interacting with ORCs, Cdc6, Cdt1 and MCM proteins [42]. MYC also stimulates the activity of the Cdc45/Mcm2-7/GINS (CMG) complex, a multi-subunit replicative helicase that associates with the pre-RC and, once activated, starts unwinding replication origins. Here, MYC promotes recruitment of the CMG complex to a subset of licensed origins proximal to MYC-bound genomic sites [43,44]. Epigenetic modifications of these loci, such as increased histone H3K4 methylation and histone hyper-acetylation, and perhaps nucleosome remodeling, could be the trigger, as suggested by studies on the well-defined *lamin B2* origin [45]. Indeed, cell cycle-dependent epigenetic changes at the E-box of the *lamin B2* gene favor the recruitment of MCM proteins and MYC-dependent replication initiation when cells enter S-phase [45]. Whether all MYC-bound loci should be considered bona fide replication origins is a matter that will deserve further investigation.

MYC-dependent control of DNA synthesis also raises the question of whether MYC, in physiological or pathological settings, may also control, or affect, the processivity and fidelity of replication, thus causing replicative stress (RS). In general, numerous events can interfere with the replication machinery, determining the slowing or the stalling of replication fork progression during DNA synthesis, thus generating replication fork collapse, DNA damage and replicative stress (Figure 2) [46]. One of the most commonly recognized RS sources is unrepaired DNA lesions generated by different types of insults, exogenous and endogenous, such as UV and ionizing radiations, chemical mutagens and by-products of cellular metabolisms, such as reactive aldehydes or reactive oxygen species (ROS). All these lesions can physically act as barriers to replication fork progression [47]. Moreover, limitation of the essential replication factors, such as nucleotides or components of the replication machinery, can slow down or stall the replication fork, thus inducing RS. RS could also be generated by improper control of replication initiation, as the firing of a too-large number of origins could deplete nucleotide pools, while the firing of too few origins could lead to under replication and loss of genetic information [47].

Additional potential sources of RS are the secondary DNA structures that can form at specific loci, such as telomeres, centromeres, ribosomal DNA loci and fragile sites [48,49,50], and the physiological obstacles presented by the transcriptional machinery [51].

Pathologically, a significant cause of replicative stress is the activation of oncogenes (oncogene-induced replicative stress), which depending on the context, may serve as either a tumor-suppressive mechanism or as a way to fuel genomic instability [52].

The mechanisms through which oncogene activation leads to replication stress include enforced cell cycle regulation, impairment of origin licensing and/or origin firing, defects in nucleotide metabolism and conflicts between DNA replication and transcription [52]. Oncogene activation can perturb replication dynamics by lack of origin activation either due to a reduction in the number of activated origins in early S-phase, as is the case for cyclin E overexpression [53], or through direct inhibition of origin firing, as shown for the *Mdm2* oncogene [54]. On the other hand, oncogene activation can induce replication stress by hyper-activation of origins due to the overexpression of licensing factors [55], the induction of proliferation by forcing G1/S transition, by indirectly causing a reduction in replication fork rate [56] or by promoting re-replication [57,58].

Metabolic control of DNA synthesis is frequently a limiting factor in oncogene-induced proliferation, leading to depletion of the available nucleotide pools. This is the case of oncogenes such as Cyclin E and HPV E6/E7, which promote S-phase without inducing nucleotide biogenesis, and provoke potent RS due to insufficient levels of dNTPs. In those instances, RS could be rescued by either metabolic supplementation or MYC expression [59]. Similarly, deceleration of replication forks induced by oncogenic Ras was also rescued by supplementing nucleoside [60]. Deficiency of folate, which is essential for dNTP biosynthesis, enhanced Ras- and Cyclin E-induced replication stress and tumorigenesis in mice [61]. In some instances, oncogenes may directly interfere with nucleotide biosynthesis [62], as in the case of BCL2, which inhibits the ribonuclease reductase complex, thus causing dNTP pool depletion and premature termination of replication forks [63].

MYC’s ability to induce RS has been historically linked to the up-regulation of CDKs and E2Fs, which are known to alter cell cycle dynamics and to accelerate S-phase entry [35]. Yet, it should be noted that MYC is less proficient than other classical oncogenes in inducing RS, possibly owing to its pleiotropic control of several processes linked to DNA replication (Figure 2).

Initial evidence linking MYC to the induction of replicative stress began to emerge in the late 1980s, when a direct role of MYC overexpression in DNA replication started to emerge [39,41]. In the following years, different groups showed that MYC deregulation could lead to endoreduplication [64,65,66,67,68,69,70,71] and that MYC was able to promote re-replication in specific loci and chromosomal regions by acting as an illegitimate replication-licensing factor [72].

Over the years, it has become increasingly evident that MYC may also induce RS independently of its transcriptional activity. In fact, MYC overexpression leads to increased and premature origin firing, linked to origins’ hyper-activation, while its depletion decreases the number of active origins [42,43,46,73]. Moreover, MYC interaction with the pre-replicative complex alters the spatiotemporal program of replication initiation and results in elevated replication fork stalling or collapse, thus causing DNA damage [42,43,73].

Genome-wide mapping of MYC-induced DNA replication revealed that anticipated S-phase entry, which is a consequence of MYC overexpression, is associated with the firing of a novel class of origins (oncogene-induced origins) that are preferentially located near genes normally transcribed during G1. These origins, which are proximal to highly expressed MYC targets, are prone to premature termination and are a potential source of genomic instability, possibly resulting from conflicts between DNA replication and the transcriptional machinery [74]. Thus, it is reasonable to hypothesize that MYC-driven hyper-transcription might predispose cells to replication stress, as argued in the following paragraph.

## 7. Linking MYC-Dependent Transcription to Replicative Stress

It is generally accepted that oncogene-dependent hyper-transcription may lead to replicative stress [52,60,75,76,77]. This has been linked to the formation of paused transcriptional complexes, which by forming stable DNA–RNA hybrids and R-loops, pose a physical and topological challenge to the DNA replication machinery, slowing down and stalling DNA replication forks [52,60]. These conflicts are resolved by the generation of DNA double-strand breaks. The observation that oncogene-induced replicative stress can be phenocopied by the over-activation of a general transcription factor, such as TBP, formally supports the hypothesis that hyper-transcription can be the causal trigger [60].

Even though this evidence establishes a direct link between robust transcriptional activation and transcription-dependent replicative stress (TDRS), oncogenic MYC triggers mild replicative stress.

At first glance, this appears paradoxical since oncogenic activation of MYC leads to selective transcriptional amplification of its targets [4,8]. In addition, the anticipated S-phase entry and the firing of oncogene-induced replication origins, which occurs upon MYC overexpression [74], are expected to set the ideal stage for TDRS. A possible explanation for this paradox could be the presence of compensatory activities, which may improve robustness and processivity to MYC-dependent transcription, thus attenuating both transcriptional stress and TDRS. A corollary of this hypothesis is that the lack of MYC-induced transcriptional stress would relieve the need for genetic adaptation during tumor development, thus leaving MYC-dependent tumors liable to therapeutic strategies that evoke TDRS.

While still understudied, there is emerging evidence supporting that, in MYC overexpressing cells, selective modulation of pathways and genes linked to transcriptional control can exacerbate latent transcriptional stress (Figure 1). Interesting observations come from the study of INO80c, a complex which remodels chromatin by repositioning, spacing, evicting and exchanging histones [78]. INO80c prevents transcriptional and replicative stress responses by facilitating replication fork restart after stalling and by supporting the removal of RNA pol II at transcribed genes following the collision of a replication fork with transcription. INO80c also promotes general eviction and degradation of nucleosomes following high doses of oxidative DNA damage and represses anti-sense transcription at promoters [78].

TIP49 (RUVBL1), an ATP-dependent helicase which is a component of the INO80c, physically interacts with MYC and is required for MYC-dependent transformation, but is dispensable for the growth of immortalized cells [21]. Recent studies indicated that TIP49 regulates the release of promoter-paused RNA pol II; while its overexpression favored RNA pol II escape from promoters, its silencing led to increased pausing. Interestingly, in both instances, there was the induction of replicative stress. These results may highlight the requirement for coordinated control of RNA pol II dynamics during DNA replication [79] and may suggest that recruiting INO80c might balance RNA pol II pausing in MYC-transformed cells, in order to prevent transcription-dependent replicative stress.

Along the same lines, recent studies have identified a synthetic lethal interaction between MYC and UVSSA [80]. UVSSA is an adaptor protein, part of the CSA(ERCC8)/UVSSA/USP7 complex, a key component of the transcription-coupled repair pathway (TCR) [81]. This complex is co-opted when elongating RNA pol II stalls following the encounter of bulky DNA adducts. The CSA(ERCC8)/UVSSA/USP7 complex contributes to the stabilization of the CSB protein [81], controls ubiquitination of stalled RNA pol II on K1268 [82,83] and allows the recruitment of TFIIH [84]. All these events are needed for DNA repair and proper restart of transcription at damaged sites. In cells where MYC activity was elevated, the loss of UVSSA, as well as of the other components of the complex, led to decreased stalling of RNA pol II on paused promoters, activation of the DNA damage response and loss of fitness [80]. Overall, this suggests that the absence of UVSSA leads to the induction of transcriptional stress and genome destabilization. Given that the CSA(ERCC8)/UVSSA/USP7 complex promotes ubiquitination of stalled RNA pol II, participates in the regeneration of hypo-phosphorylated RNA pol II complexes during TCR and recruits TFIIH [83,84], it is conceivable that this complex may also be crucial under conditions of aberrant RNA pol II regulation, such as during MYC overexpression. Further work may elucidate whether bulky DNA adducts or topological intermediates may lead to RNA pol II stalling on MYC target genes, establish if UVSSA is recruited at these sites, and explain how transcriptional dynamics are affected by UVSSA.

Further insight comes from studies of the MYCN paralogue, mainly from the Eilers’ laboratory. They showed that S-phase transcription relies on the activity of the Aurora-A kinase, which is recruited on MYCN-regulated promoters to support proper post-replicative deposition of the histone H3.3 variant at the +1 position of the transcription start site. Once recruited on these promoters, Aurora-A phosphorylates the Ser10 residue of Histone 3, promoting H3.3 incorporation onto chromatin and its correct positioning, which allows nucleosome depletion at the transcriptional start site. Inhibition of Aurora-A led to the loss of nucleosome remodeling, which caused pronounced accumulation of elongating RNA pol II in promoter-proximal regions indicative of abortive and stalled elongation. This was associated with the formation of RNA–DNA hybrids which inhibited early termination by RNA decapping enzymes. Consequently, the lack of RNA pol II clearance led to replicative stress [85].

The identification of a critical protective function of BRCA1 in controlling MYCN-dependent transcription reinforces the concept that clearance of stalled RNA pol II is a critical step to prevent transcriptional and replicative stress during S-phase [86]. In MYCN-amplified neuroblastoma cells, S- and G2-phase-dependent transcription is safeguarded by the MYCN-dependent recruitment of USP11 and BRCA1. This trimeric complex promotes early transcriptional termination by favoring nascent RNA decapping of poised RNA pol II, thus allowing its release from pausing sites. Clearance of poised RNA pol II transcriptional complexes is needed to relieve the torsional stress accumulated at promoter-proximal regions which is associated with R-loop accumulation [86].

Interestingly, loss of BRCA1 did not lead to increased genome instability or DNA damage [86], suggesting that transcriptional stress might be particularly genotoxic when associated with replication defects or, possibly, that rescue pathways are in place to avoid the genotoxic consequences of stalled transcriptional complexes. While this function of BRCA1 does not seem to be regulating similar processes on MYC target genes [86], it is possible that orthologue pathways may serve a similar purpose.

## 8. Replicative Stress Is a Therapeutic Target for MYC-Driven Cancers

Oncogene-induced RS is one of the main sources of genomic instability that can lay the foundations for tumor evolution. On the other hand, it can also act as an anticancer barrier due to its tumor-suppressive activity at the early stages of cancer development [47,58,87]. Typically, this imposes a strong selective pressure to genetically bypass such tumor-suppressive responses, which are induced by oncogenes during tumor progression. On the contrary, MYC-induced tumors may lack such genetic selection, since they display attenuated RS. This peculiarity represents an attractive liability that could be exploited therapeutically, by specifically eliciting acute replicative DNA damage in cancer cells. Thus, molecular understanding of the mechanisms suppressing inherent RS is a fundamental pre-requisite to the implementation of preclinical strategies based on the induction of RS in MYC-driven tumors.

Growing evidence suggests that MYC mitigates RS by controlling processes related to nucleotide biosynthesis, DNA replication, checkpoint control and DNA repair (Figure 3). More recent data also suggest that the fine tuning of RNA polymerase dynamics may prevent transcription-dependent RS (as presented in the preceding paragraph).

As mentioned before, oncogene activation may interfere with nucleotide pools, typically leading to dNTP depletion. By contrast, oncogenic activation of MYC induces the expression of several genes involved in nucleotide biosynthesis, increasing dNTP pools, enabling cell proliferation [37,59,88] and bypassing RS due to high rates of DNA synthesis [59].

Mitigation of RS can also be linked to a housekeeping function of MYC in maintaining genome integrity. MYC binds the promoter region of several DSB repair-related genes, such as *Ku70, Rad51, BRAC2*, *Rad50* and *DNA-PKcs*, inducing their transcription [89]. Coherently, MYC silencing leads to the decrease in ATM and DNA-PKcs, which, in turn, results in reduced DSB repair [90]. More recently, proper turnover of promoters’ bound MYC was shown to be required for the efficient repair of TSS-associated DSBs [26].

Suppression of RS can be also achieved by the MYC-dependent up-regulation of enzymes involved in DNA replication, such as the MRN complex (MRE11/RAD50/NBS1) which mediates DNA repair by homologous recombination, is necessary for the restart of collapsed forks [91] and confers tolerance to transcription–replication conflicts [92,93]. Similarly, the WRN helicase, which is involved in the resolution of unusual replication intermediates, is required to repress MYC-induced RS [25,94,95].

Other factors, such as the fork-remodeling proteins Smarcal1 and Zranb3, have been shown to stabilize replication forks and to prevent fork collapse during MYC-induced transformation [96]. Smarcal1 and Zranb3, which belong to the SWI/SNF helicase family, protect the reversed fork from nucleolytic degradation. They can remodel stalled replication forks by both fork reversal and fork restoration, and display different substrate specificities which depend both on the structure of the fork junction and the proteins associated with the stalled fork [97].

Along the same lines, Polη, a translesion synthesis polymerase (TLS), suppresses MYC-induced RS. Beyond canonical TLS, this polymerase participates in homologous recombination, nucleotide excision repair, re-replication and replication of ‘hard-to-replicate’ regions. In MYC-overexpressing cells, Polη is recruited to replication foci by CDC45- and RAD18-mediated ubiquitylation of PCNA, and its loss generates cell cycle arrest, fork stalling and DSBs. This suggests that MYC-induced DNA replication may require dedicated DNA replication mechanisms to overcome intrinsic RS [98].

Another example of how MYC tempers inherent RS is reported in breast cancer stem cells (CSCs), where the collision between transcription and replication machineries is frequent and the major cause of RS in tumor spheroids [99]. Here, it was shown that MYC-induced expression of MCM10 is critical for the maintenance of breast CSCs and that MCM10 may buffer RS by activating dormant origins to ensure replication progression [99].

The control of the 3D architecture of DNA replication and possibly chromatin looping [100] are likely key factors in preventing RS. In this regard, cohesins, by their coordinated control of DNA repair [101], DNA synthesis [102] and transcription [103], may play a relevant role. Indeed, silencing of the cohesin complex component RAD21 affected MYC-dependent transcription and altered replication forks [104]. Interestingly, the forced activation of MYC in RAD21-depleted cells restored MYC-dependent transcription and S-phase entry but triggered defective replication fork initiation and progression, leading to acute RS. These observations may suggest that cohesins and proper control of cohesion are required for faithful DNA replication and to prevent MYC-induced replication stress [104].

Altogether, this evidence highlights a wide array of processes that counterbalance MYC-induced RS, and provides the rational for the design of therapeutic strategies aimed at evoking toxic RS in MYC-induced tumors.

Most of these studies have so far focused on the targeting of the ATR/CHK1 pathway, with the idea that latent RS in MYC-dependent tumor cells might constantly engage ATR/CHK1 to prevent rampant genomic instability. The ATR/CHK1 pathway controls cell cycle checkpoints, stability of stalled replication (to prevent their collapse and the formation of DSBs) and supports the restart of stalled replicative complexes. When RS ensues, fork stalling would generate persistent ssDNA, which creates a platform for the recruitment of upstream DDR factors such as RPA, ATR and CHK1, and the subsequent activation of the signal transduction chain leading to cell cycle arrest, stabilization or resolution of the stalled fork and the restart of DNA replication [46]. Consequently, the ATR/CHK1 pathway represents a vulnerability for tumors in which latent RS can be awakened [105] and, consistently, MYC-driven cancers are highly sensitive to ATR and CHK1 inhibitors as single agents [106,107,108,109]. While this synthetic lethality has been widely confirmed by several studies, the molecular cause(s) are still ill defined.

In some cases, concomitant inhibition of both CHK1 and ATR improves therapeutic efficacy as in high-risk medulloblastomas, where high MYC levels were associated with hypersensitivity to pharmacological inhibition of either CHK1 or ATR. Intriguingly, this approach also potentiates the anticancer efficacy of cisplatin and is well tolerated in non-cancerous neuronal cells [110]. The liaison between MYC and ATR/CHK1 may extend beyond tumors bearing genomic rearrangements of the MYC locus; in non-small cell lung cancer (NSCLC) the loss of the tumor suppressor PPP2R2A, which inhibits MYC translation, leads to high MYC activity and determines sensitivity to ATR or CHK1 inhibitors due to acute RS responses [111]. This also exemplifies how knowledge of the mechanisms leading to MYC deregulation might be used to identify biomarkers for therapy response.

Importantly, ATR/CHK1 targeting can also be exploited in combinatorial regimens. In multiple myeloma, ATR activity is necessary to compensate MYC-induced RS and its inhibition triggers apoptosis [112]. When ATR inhibition is combined with an oxygen-radicals inducer, such as piperlongumine, it is possible to selectively target the subset of multiple myeloma presenting high levels of RS. The possibility to concomitantly enhance oxidative stress and to block RS response might open up the possibility to treat multiple myeloma patients with aggressive disease and poor prognosis. In non-germinal center B-cell diffuse large B-cell lymphoma (GBC-DLBCL), characterized by high MYC protein expression and *CDKN2A/B* deletion, concomitant inhibition of ATR and the WEE1 kinase leads to DNA damage and G1 and intra S-phase cell cycle arrest, which pair with a strong cytotoxic response in in vivo DLBCL models [113].

Combined ATR and Aurora-A inhibition demonstrated selective efficacy in a mouse model of MYCN-derived neuroblastoma, leading to permanent eradication of tumors in a subset of mice [85]. Here, ATR inhibition exacerbated the cytotoxic effect of transcription–replication conflicts generated by Aurora-A inhibition.

Other approaches to evoke RS rely on the metabolic targeting of nucleotide biosynthesis pathways. In MYC-driven medulloblastoma, the combination of AZD1775, an inhibitor of the cell cycle checkpoint kinase WEE1, with gemcitabine, an antimetabolite that attenuates nucleotide synthesis and promotes replicative DNA damage, significantly increased cytotoxic DNA damage and resulted in full regression of murine tumor xenografts [114]. In MYC-mediated T-cells acute lymphoblastic leukemia, CHK1 inhibition impaired cell viability and induced a remarkable synthetic lethality with the mTOR inhibitor rapamycin [115]. Such synergy was due to the inactivation of carbamoyl phosphate synthetase 2, aspartate transcarbamoylase and dihydroorotase (CAD), the essential enzymes for the three steps of de novo pyrimidine synthesis. This boosted MYC-induced RS and enhanced the anti-tumoral efficacy of CHK1 inhibitors.

Finally, there is genetic and pharmacological evidence suggesting that MYC-induced RS may be evoked by targeting DNA repair or fork stability pathways. Several reports indicate sensitization of MYC-induced tumors to PARP inhibitors [116,117,118,119]. In particular, in neuroblastoma models, this has been linked to enhanced RS. High-risk, MYC-amplified or MYC-overexpressing neuroblastomas present poor prognosis and show elevated expression of PARP-1 and PARP-2. Preclinical therapies based on PARP inhibitors exacerbate replicative stress, by enhancing fork stalling, DDR, cell cycle arrest and post-mitotic cell death [120,121]. The role of MYC as a sensitizer to PARPi-based therapy is also reported in patient-derived glioblastoma stem-like cells (GSCs), which show MYC or MYCN amplification and are more sensitized to PARPi when ATR activity is pharmacologically inhibited [122]. This is linked to MYC-dependent repression of *CDK18,* a cyclin-dependent kinase which associates with ATR-containing complexes involved in DNA repair and S-phase progression. MYC-induced repression of *CDK18*, as well as ATR inhibition, affects ATR activity, and increases sensitivity to PARPi, thus representing a promising therapy for glioblastoma and, potentially, other PARPi-refractory tumors.

Emerging data also suggest that MRE11 targeting could be appropriate to treat MYCN-driven cancers. MRE11 is essential to prevent deleterious accumulation of DNA damage and its increased expression in MYCN-amplified neuroblastoma is required to restrain MYCN-dependent RS [91]. Pharmacological inhibition of MRE11, through encapsulation of the mirin drug in nanoparticles, induces accumulation of RS, impairment of tumor growth and apoptosis in MYCN-amplified neuroblastoma xenografts in vivo. Furthermore conditional deletion of *MRE11* in B-lymphocytes prevented the development of *IgH–MYC* translocated lymphomas and triggered conspicuous DDR responses, suggesting that MRE11 counteracts MYC-induced RS [123].

Concerning DNA replicative complexes, proof of principle preclinical genetics suggests that depletion of MCM10 protein induces RS in breast CSCs, but not in the normal counterpart, thus causing the demise of cancer cells. As breast CSCs are resistant to paclitaxel and other chemotherapeutics, targeting the MYC–MCM10 axis has been proposed as a therapeutic strategy for preventing drug resistance and recurrence of breast tumors [99]. Finally, genetic analysis of components of the SNF2 family of DNA translocases revealed an essential role for Zranb3 in controlling fork progression and stability in MYC-driven B-cell lymphomas [96]. Loss of Zranb3 led to sharp increase in DNA damage and apoptosis in pre-tumoral B-cells, thus causing a significant delay in lymphoma development. Inhibition of Zranb3, alone or in combination with replication stress-inducing drugs, could provide a therapeutic benefit not only in MYC-driven lymphomas but also for other MYC-driven malignancies [96].

## 9. Conclusions and Perspectives

Overall, there is increasing understanding of how MYC controls gene expression and genome replication by sitting at the crossroad of these processes. Yet, key challenges are lying ahead and further work will be needed to fully describe and appreciate the intricacy of such regulation. While it is clear how MYC, by controlling expression of replication factors, metabolic enzymes and checkpoint components, contributes to high-fidelity DNA replication, additional work will be needed to fully appreciate its role in this process. Indeed, direct interaction of MYC with replicative factors and its interaction with the pre-replicative complex calls for the need of parsing out what might be MYC contribution to DNA replication per se, and whether such interactions might also reflect the proximity of MYC-directed transcription to (early) replicating regions. Genome-wide chromatin association studies and 3D nuclear architecture analysis will be pivotal to understand how chromatin looping and nuclear organization may prevent, or predispose, transcription–replication conflicts. Along the same lines, topological aspects of replication and transcription will need to be clarified, in particular concerning the role of cohesins.

Clearly, the regulation of RNA polymerase dynamics is central to temper transcriptional and replicative stress. Additional detailing of how polymerase stalling and pausing might be resolved, not only at promoter-proximal regions, but also at gene bodies and at termination sites, will complement current knowledge and may offer additional mechanistic clues.

Still poorly explored is how co-transcriptional RNA processing, by affecting RNA pol II dynamics, may generate RS. Evidence of the synthetic lethality of splicing defects with MYC overexpression [124,125], and the observation that the bypass of early termination alters splicing [126], raise the possibility that unresolved co-transcriptional splicing intermediates, if not properly processed and cleared, may increase the chances of replicative conflicts.

In conclusion, a detailed understanding of the molecular mechanisms mitigating MYC-induced replication stress, together with the characterization of all the factors involved in these processes, will be a strategic asset to identify potential therapeutic targets for the treatment of MYC-driven cancers. 

## Figures and Tables

**Figure 1 ijms-22-06168-f001:**
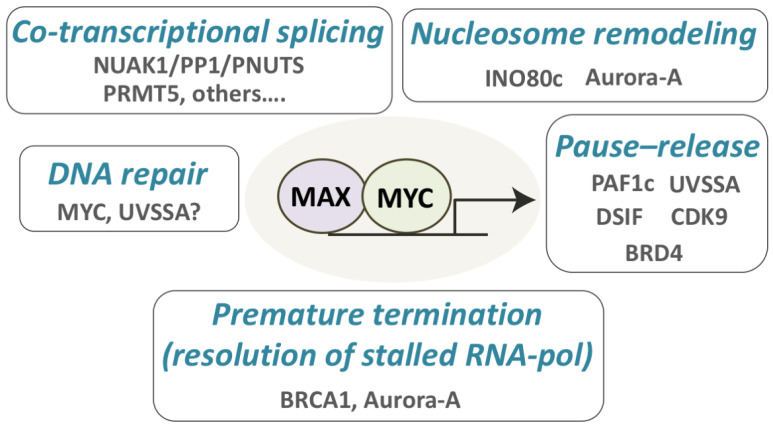
Processes and factors that contribute to MYC-induced hyper-transcription and are needed to prevent transcriptional stress.

**Figure 2 ijms-22-06168-f002:**
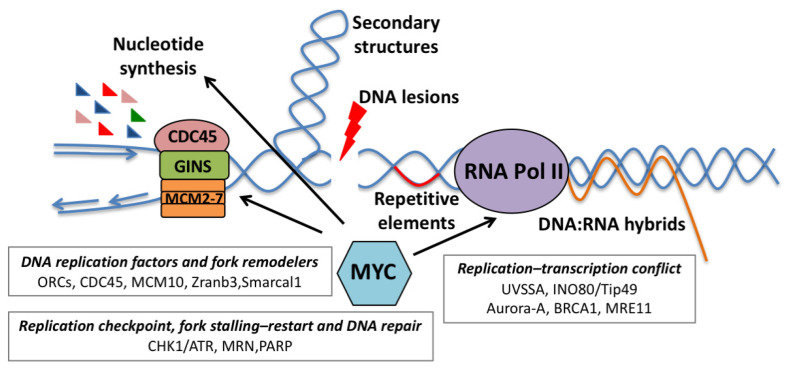
Schematic view of the most common mechanisms leading to replicative stress. Arrows are pointing to those events that are controlled by MYC and that can either support DNA replication or can predispose cells to replicative stress. Text boxes: details for the processes and the factors implicated in MYC-induced replicative stress.

**Figure 3 ijms-22-06168-f003:**
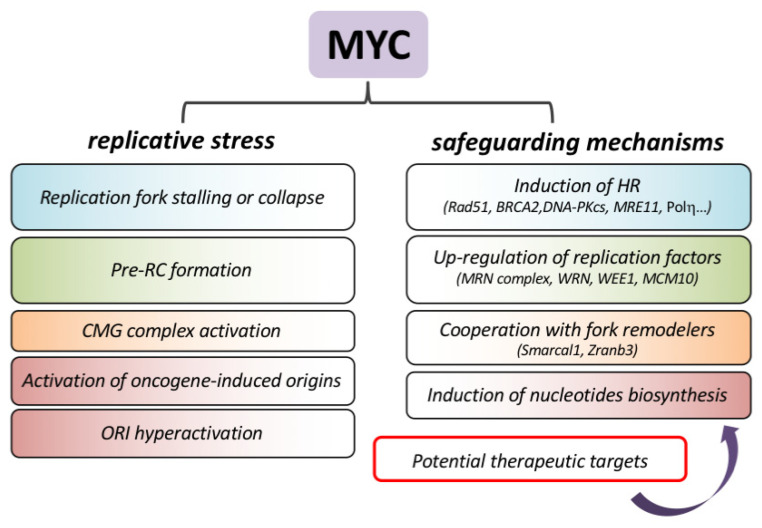
Overview of potential sources of MYC-induced replicative stress (on the left) and of processes that render replication stress latent in MYC-overexpressing cells (on the right). As argued in the text, these processes represent an attractive liability of MYC-driven cancers that could be exploited therapeutically.

## Data Availability

Not applicable.

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
