# Peer review of "MYC-Induced Replicative Stress: A Double-Edged Sword for Cancer Development and Treatment"

_ijms, 2021, doi:10.3390/ijms22126168_

Round 1

Reviewer 1 Report

This is a well written and clear review/perspective that introduces the relevant parts of the complex field of MYC and its activities and interactors, before focussing on replication stress and how RS might provide a therapeutic target, especially in MYC driven tumors.

I only have a few minor points that would benefit from correction or greater explanation:

  1. Line 31, should be “growth factor signaling”.
  2. Not sure what is meant by the word “tonic” on line 47 - -“constitutive”???
  3. Line 88, should be “when MYC levels are raised”.
  4. Line 100 “studies have resulted in better structure-function” would be clearer.
  5. Line 141 and elsewhere, the convention is generally RNA pol I, RNA pol II and RNA pol III rather than 1,2 & 3 – unless the journal has a specific style requirement in this respect.
  6. Line 146, I think the term non-canonical PRC1 complex needs some explanation as many potential readers may not be familiar with this.
  7. Line 149, it is not clear to me what “cotranscriptional processing of MYC regulated genes” is referring to and how this links to the previous clause – need rewording or a bit more explanation.
  8. Line 186, I had to look up what “propedeutic” means, would something like “is required for” be better?
  9. Lines 212-13, a bit more explanation is required to link with the first part of this sentence.
  10. Line 452, it is probably relevant and important to mention that NBS is also involved in homologous recombination repair, which is essential for replication fork restarting.

Author Response

We would like to thank the reviewer for the positive and constructive comments on our manuscript. Following we are reporting a point-by-point response where we did our best to comply with their requests and suggestions. We should point out that we extensively revised and implemented chapter 8 to improve the clarity and include some articles that were omitted in the previous version.

Reviewer 1

This is a well-written and clear review/perspective that introduces the relevant parts of the complex field of MYC and its activities and interactors, before focusing on replication stress and how RS might provide a therapeutic target, especially in MYC driven tumors.

I only have a few minor points that would benefit from correction or greater explanation:

1. Line 31, should be “growth factor signaling”.

The text was amended as suggested, keeping the US spelling for coherence (signaling vs signaling).

2. Not sure what is meant by the word “tonic” on line 47 - -“constitutive”???

Tonic was intended as a synonim for constitutive. We have revised the text as suggested.

3. Line 88, should be “when MYC levels are raised”.

Text was amended as suggested

4. Line 100 “studies have resulted in better structure-function” would be clearer.

We propose the following: “Several biochemical studies have contributed to the structure-function insight on MYC and the MYC/MAX heterodimer” (lines 101-102)

5. Line 141 and elsewhere, the convention is generally RNA pol I, RNA pol II and RNA pol III rather than 1,2 & 3 – unless the journal has a specific style requirement in this respect.

Text was amended as suggested

6. Line 146, I think the term non-canonical PRC1 complex need some explanation as many potential readers may not be familiar with this.

Text was amended as follows:” non-canonical PRC1 components (i.e. co-factors that associate to the core components of the PRC1 complex)” (lines 148-149)

7. Line 149, it is not clear to me what “cotranscriptional processing of MYC regulated genes” is referring to and how this links to the previous clause – need rewording or a bit more explanation.

Here we intended to suggest that the interaction of MYC with RNA processing factors and RNA binding proteins may imply that MYC is not only controlling transcription, but may also co-ordinate downstream events like splicing or the interaction of nascent RNA with RNA binding proteins.

We have tried to make the text clearer by re-phrasing as follows:

The same study showed MYC interaction with several RNA processing factors and spliceosomal components, as well as with proteins involved in ribosome biogenesis. This perhaps suggests that MYC may regulate co-transcriptional processing of its target genes to regulate splicing of nascent RNA or to foster the inclusion of newly transcribed RNA into ribonucleoprotein complexes.” (lines 150-155)

8. Line 186, I had to look up what “propedeutic” means, would something like “is required for” be better?

Yes, we amended as suggested by the reviewer.

9. Lines 212-13, a bit more explanation is required to link with the first part of this sentence.

We tried to rephrase this sentence in order to make more explicit the connection between the association with epigenetic complexes known to repress transcription by altering chromatin post translational modifications and the inferred mechanism of MYC dependent repression. It reads “The identification of the interaction of MYC with histone deacetylase complexes (SIN3a and BHC) and non-canonical components of the PRC-1 complex may suggest that, at least in some cases, MYC repress transcription by epigenetic mechanisms, thus favoring the removal of activating histones modifications (i.e acetylation of histones) or the deposition repressive chromatin marks (i.e. monoubiquitylation of histone H2A at lysine 119) [18].” (LINES 216-221)

10. Line 452, it is probably relevant and important to mention that NBS is also involved in homologous recombination repair, which is essential for replication fork restarting.

As suggested we clarified the role of the MRN complex in DNA repair and fork restart, and added new references. The sentence now reads:  “Suppression of RS can be also achieved by the MYC dependent up-regulation of enzymes involved in DNA replication, such as the MRN complex (MRE11/RAD50/NBS1) which mediates DNA repair by homologous recombination, is necessary for the restart of collapsed forks[92], and confers tolerance to transcription-replication conflicts[93, 94].” (lines 456-459).

Reviewer 2 Report

This a nice, comprehensive and valuable review. The paper reviews the induction of the replicative stress by MYC. As rightly pointed out by the authors, this MYC activity could be targetd in tumor therapy

This is a critical MYC role and to my knowledge, it has not been appropriately dealt with in such a extended and comprehensive review.  The authors have made relevant contributions in the field of the transcriptional activity of MYC over the recent years

I have some comments:

1.-A big deal of the review deals with the transcriptional activity of MYC, indeed the five first sections of the review are focused in that. Whether or not MYC acts as a general amplifier of transcription or a more or less specific transcripton factor has been debated over the last years. In the page 3 the review gives a good state-of-the art  of this controversial issue, and the the author's model reconciles the data in hand and is  well explained. I wonder whether this might be reinforced with a figure

2.- Page 4, lines 175. The author describe the onteraction of MYC with TBP. However, about half of human promoters lack TATA box but have a GpC island at rhe promoter. Can the authors may comment on how MYC activates these genes?

3.-Page5. Section 6. In this section, although it is probably not strictly required,  I miss the mentioning of original research papers showing that MYC induces "illegitimate" DNA replication resulting in G2 accumulation and/or endoreplication, both markers of replicative stress, I have listed below some of these papers (there are probably more) 

Gatti G, Maresca G, Natoli M, Florenzano F, Nicolin A, Felsani A, et al. MYC prevents apoptosis and enhances endoreduplication induced by paclitaxel. PLoS One 2009;4:e5442.

 Sheen JH, Dickson RB. Overexpression of c-Myc alters G(1)/S arrest following ionizing radiation. Mol Cell Biol 2002;22:1819–33.

Albajar M, Gómez-Casares MT, Llorca J, Mauleon I, Vaqué JP, Acosta JC, et al, MYC in chronic myeloid leukemia: induction of aberrant DNA synthesis and association with poor response to imatinib. Mol Cancer Res. 2011;9:564-76.

Cowling VH, Chandriani S, Whitfield ML, Cole MD. A conserved Myc protein domain, MBIV, regulates DNA binding, apoptosis, transformation, and G2 arrest. Mol Cell Biol 2006; 26:4226-39.

Felsher DW, Zetterberg A, Zhu J, Tlsty T, Bishop JM. Overexpression of MYC causes p53-dependent G2 arrest of normal fibroblasts. Prc Natl Acad Sci USA 2000 Sep 12;97:10544-8.

-Page 6, line 260: “Whether all MYC bound loci should be considered bona fide replication origins is a matter that will deserve further investigation. THis is a good point Do the authors mean that all MYC bound sites  may eventually work as replication origins?

Page 8, line 365. The  relationship between MYC and INO80c/TIP49 can be included in Figure 2?

-Page 9. “Section 8. Replicative stress is a therapeutic target for MYC-driven cancer”. In general this paragraph is difficult to follow. Maybe it can be arranged so the reader can clearly see which processes depend on the interaction of MYC with a replicative or repair protein and which depends on the transcriptional activation of replication- or DNA repair genes by MYC. Also , it is possible to include in Figure 2 more o the interactions mentioned n the text?

Page 9, line 462-464: This paragraph is not directly linked to the topic of the review. If that is the case,  please explain

Page 10, line 476. The impact of MYC on th synthetic lethality between ATR and Chk1 inhibition is not clear in ths paragraph. Can this be better explained? 

Page 10-lines 500: please give some information on the zinc finger protein Zranb3

Typos in  page 5, lines 215-18: MYZ1 instead of MIZ1

Author Response

We would like to thank the reviewers for the positive and constructive comments on our manuscript. Following we are reporting a point by point rebuttal where we did our best to comply with their requests and suggestions. We should point-out that we extensively revised and implemented chapter to improve the clarity and include some articles that were omitted in the previous version.

Reviewer 2

This a nice, comprehensive and valuable review. The paper reviews the induction of the replicative stress by MYC. As rightly pointed out by the authors, this MYC activity could be targeted in tumor therapy

This is a critical MYC role and to my knowledge, it has not been appropriately dealt with in such an extended and comprehensive review.  The authors have made relevant contributions in the field of the transcriptional activity of MYC over the recent years.

I have some comments:

1.-A big deal of the review deals with the transcriptional activity of MYC, indeed the five first sections of the review are focused in that. Whether or not MYC acts as a general amplifier of transcription or a more or less specific transcription factor has been debated over the last years. In the page 3 the review gives a good state-of-the art  of this controversial issue, and the author's model reconciles the data in hand and is  well explained. I wonder whether this might be reinforced with a figure

As the reviewer pointed out, we gave a detailed account of how MYC controls transcription. We felt this extensive background was needed in order for the general reader to be able to fully appreciate the content of chapter 7, where we discuss transcription dependent replicative stress. We thank the reviewer for the kind appreciation of our effort to clarify the issue of transcriptional amplification vs selective control and apology for our lack of creativity which prevented us from coming up with a graphical summary of this (and is the reason why we were not able to comply with the suggestion of the reviewer of adding one figure on this issue).

2.- Page 4, lines 175. The author describe the Interaction of MYC with TBP. However, about half of human promoters lack TATA box but have a GpC island at the promoter. Can the authors may comment on how MYC activates these genes?

The reviewer is pointing one of the outstanding questions in the field which is to understand the molecular details of how MYC controls transcription from promoters of gene that may be structurally different. Indeed as pointed out by the reviewer a consistent fraction of MYC regulated genes is likely to lack a TATA-box (to my knowledge there is no published analysis on this). Considering that there is general agreement in the field on the idea that transcription from TATA-less promoters is still dependent on TBP and its associated factors (TAFs), it is reasonable to speculate that MYC might serve as a recruiter for TBP on TATA-less promoters. We have added a sentence to clarify this: “Of note, this protein-protein interaction between MYC and TBP may suggest a mechanism for TBP recruitment on TATA-less MYC targets.” (lines 185-186) We guess that addressing this issue would be a reasonable follow-up of the elegant structure-function analyses recently published by the Penn & Sunnerhagen laboratory.

3.-Page5. Section 6. In this section, although it is probably not strictly required,  I miss the mentioning of original research papers showing that MYC induces "illegitimate" DNA replication resulting in G2 accumulation and/or endoreplication, both markers of replicative stress, I have listed below some of these papers (there are probably more) 

Gatti G, Maresca G, Natoli M, Florenzano F, Nicolin A, Felsani A, et al. MYC prevents apoptosis and enhances endoreduplication induced by paclitaxel. PLoS One 2009;4:e5442.

 Sheen JH, Dickson RB. Overexpression of c-Myc alters G(1)/S arrest following ionizing radiation. Mol Cell Biol 2002;22:1819–33.

Albajar M, Gómez-Casares MT, Llorca J, Mauleon I, Vaqué JP, Acosta JC, et al, MYC in chronic myeloid leukemia: induction of aberrant DNA synthesis and association with poor response to imatinib. Mol Cancer Res. 2011;9:564-76.

Cowling VH, Chandriani S, Whitfield ML, Cole MD. A conserved Myc protein domain, MBIV, regulates DNA binding, apoptosis, transformation, and G2 arrest. Mol Cell Biol 2006; 26:4226-39.

Felsher DW, Zetterberg A, Zhu J, Tlsty T, Bishop JM. Overexpression of MYC causes p53-dependent G2 arrest of normal fibroblasts. Prc Natl Acad Sci USA 2000 Sep 12;97:10544-8.

We agree with the reviewer and have included in the text references of the suggested articles.

-Page 6, line 260: “Whether all MYC bound loci should be considered bona fide replication origins is a matter that will deserve further investigation. THis is a good point Do the authors mean that all MYC bound sites  may eventually work as replication origins?

Yes, given that early replication origins seem to be preferentially located on transcribed loci, that many of MYC targets are transcribed during S-phase (in particular when MYC is overexpressed) and considering the evidence that physically link MYC to the pre-replicative complex, there is mounting evidence that supports this hypothesis. Still, to our knowledge, there is no formal proof of this.

Page 8, line 365. The  relationship between MYC and INO80c/TIP49 can be included in Figure 2?

The information has been included in figure 2, as requested.

-Page 9. “Section 8. Replicative stress is a therapeutic target for MYC-driven cancer”. In general this paragraph is difficult to follow. Maybe it can be arranged so the reader can clearly see which processes depend on the interaction of MYC with a replicative or repair protein and which depends on the transcriptional activation of replication- or DNA repair genes by MYC. Also , it is possible to include in Figure 2 more o the interactions mentioned in the text?

We have extensively revised section 8, to improve the flow and the clarity of the reading. We have also included the suggested information in figure 2, as requested.

Page 9, line 462-464: This paragraph is not directly linked to the topic of the review. If that is the case,  please explain.

This paper was cited as an example of how MYC actively prevents its inherent tendency to induce replicative stress. Here the authors provide evidence that MCM10 upregulation is needed to avoid RS. The implication is that breast cancer stem cells might be eradicated by targeting those processes that are controlled by MYC and that are safeguarding DNA replication to prevent replicative stress. We have reworded the text to clarify this. (lines 476-481)

Page 10, line 476. The impact of MYC on the synthetic lethality between ATR and Chk1 inhibition is not clear in this paragraph. Can this be better explained?

ATR/CHK1 pathway has a general role in controlling cell cycle checkpoints, stabilization of stalled replication to prevent their collapse and the formation of DSBs, and supports the restart of stalled replicative complexes. The rational is that being ATR/CHK1 epistatic regulators of RS, their inhibition deploys full RS in cells which are experiencing latent. Yet, at present there is no clear indication of what is the molecular that cause make MYC overexpressing cells liable to ATR/CHK1. We have reworded the text to clarify this. (lines 495-507).

Page 10-lines 500: please give some information on the zinc finger protein Zranb3

We have reworded the text to clarify this. (lines 464-468)

Typos in  page 5, lines 215-18: MYZ1 instead of MIZ1

Amended as suggested